# A Novel Dual Fluorochrome Near-Infrared Imaging Probe for Potential Alzheimer’s Enzyme Biomarkers-BACE1 and Cathepsin D

**DOI:** 10.3390/molecules25020274

**Published:** 2020-01-09

**Authors:** Jenny M. Tam, Lee Josephson, Alexander R. Pilozzi, Xudong Huang

**Affiliations:** 1Wyss Institute and Harvard Medical School, Boston, MA 02115, USA; jenny.tam@wyss.harvard.edu; 2Center for Molecular Imaging Research (CMIR), Department of Radiology, Massachusetts General Hospital and Harvard Medical School, Charlestown, MA 02129, USA; ljosephson@mgh.harvard.edu; 3Neurochemistry Laboratory, Department of Psychiatry, Massachusetts General Hospital and Harvard Medical School, Charlestown, MA 02129, USA; APILOZZI@mgh.harvard.edu

**Keywords:** Alzheimer’s disease, BACE1, cathepsin D, biomarker, near-infrared fluorescent probe, molecular imaging

## Abstract

A molecular imaging probe to fluorescently image the β-site of the amyloid precursor protein (APP) cleaving enzyme 1 (BACE1) and cathepsin D (CatD) enzymes associated with Alzheimer’s disease (AD) was designed and synthesized. This imaging probe was built upon iron oxide nanoparticles (cross-linked dextran iron oxide nanoparticles, or CLIO). Peptide substrates containing a terminal near-infrared fluorochrome (fluorophore emitting at 775 nm for CatD or fluorophore emitting at 669 nm for BACE1) were conjugated to the CLIO nanoparticles. The CatD substrate contained a phenylalanine-phenylalanine cleavage site more specific to CatD than BACE1. The BACE1 substrate contained the sequence surrounding the leucine-asparagine cleavage site of the BACE1 found in the Swedish mutation of APP, which is more specific to BACE1 than CatD. These fluorescently-labeled peptide substrates were then conjugated to the nanoparticle. The nanoparticle probes were purified by gel filtration, and their fluorescence intensities were determined using a fluorescence plate reader. The CatD peptide substrate demonstrated a 15.5-fold increase in fluorescence when incubated with purified CatD enzyme, and the BACE1 substrate exhibited a 31.5-fold increase in fluorescence when incubated with purified BACE1 enzyme. Probe specificity was also demonstrated in the human H4 neuroglioma cells and the H4 cells stably transfected with BACE1 in which the probe monitored enzymatic cleavage. In the H4 and H4-BACE1 cells, BACE1 and active CatD activity increased, an occurrence that was reflected in enzyme expression levels as determined by immunoblotting. These results demonstrate the applicability of this probe for detecting potential Alzheimer’s enzyme biomarkers.

## 1. Introduction

The pandemics of Alzheimer’s disease (AD) and related dementias (ADRD)-frontotemporal disorders (FTD), Lewy body dementia (LBD), vascular dementia (VD), and mixed etiology dementia (MED) has incurred colossal socio–economical burden and posed a huge challenge to our healthcare system. However, there are neither efficacious treatments nor effective prevention measures available for AD and ADRD [1]. Tremendous strides have been made in developing positron emission tomography (PET) radioligands for Aβ amyloid plaques and tau tangles-AD neuropathological hallmarks, magnetic resonance imaging (MRI) methods for brain structural and vascular lesions in living individuals for AD and ADRD detections. However, Aβ (amyloid-beta) amyloid plaques and tau tangles, brain structural and vascular changes, Lewy bodies, etc., are also found in postmortem brain tissues from cognitively normal subjects [2,3]. Thus, neuropathological signs of AD, VD, FTD, or LBD only indicate neurodegeneration but not dementia. Nevertheless, biomarkers are able to play key roles in understanding etiopathogenesis of AD and ADRD, and they are also crucial to translating basic research into the clinical arena. This is because biomarkers are more closely tied to cognitive functions, and they have become essential in trials of AD-modifying therapies, and they might even serve as surrogate endpoints in dementia treatment trials and drug discovery tools applied in the dementia animal models. Current biomarkers are either invasive or expensive: cerebrospinal fluid (CSF) sampling requires a lumbar puncture [4], which many people find objectionable. MRI and PET scans are expensive, the latter involve potentially hazardous radiation exposure [5], and they are better suited to research at academic centers but not appropriate for massive use in primary care settings and real-life communities. Hence, there is an urgent and unmet medical need for imaging probes of biomarkers that can reliably distinguish normal from abnormal brain function or cognition and robustly predict or correlate with its clinical decline. As such, our long-term goal is to develop novel molecular imaging probes of relevant biomarkers for characterizing, diagnosing, and predicting outcomes in AD and ADRD.

Nevertheless, AD diagnosis is complicated by the fact that, as of yet, no definitive in vivo diagnostic tool exists for AD patients. Rather, AD diagnosis currently relies on behavior-based tests that are not specific for AD. Evidence suggests that Aβ may be a key step during AD progression [6,7,8]. This Aβ peptide, that can aggregate into plaques, is produced after the sequential cleavage of two proteases, called β- and γ-secretase, that mediate the endoproteolysis of amyloid precursor protein (APP), a type I membrane protein [9]. Thus, β-secretase cleavage is the committed step in Aβ amyloidogenesis, and cleaving enzyme 1 (BACE1) is considered as the major form of β-secretases [10]. As such, it has become one of the therapeutic targets for AD.

Along with BACE1, cathepsins, including cathepsin D (CatD), are a part of the lysosomal system, are also thought to be part of the dysfunction involved in AD [11]. Lysosomal acidification and normal proteolytic activity are found to be somewhat compromised in Alzheimer’s disease and other diseases of the central nervous system [12]. Cathepsins, that affect the production and removal of intracellular Aβ, are upregulated [13,14], disrupting the lysosomal system to ultimately increase intracellular Aβ levels [15] to a point at which Aβ is secreted extracellularly as Aβ aggregates. In a series of experiments in mice, APP degradation in vivo was mediated by several proteases whose overexpression or deletion appropriately altered Aβ levels [16,17]. CatD was shown to play a role in the processing of APP to form Aβ [18]. In these sets of experiments, Siman and co-workers used several irreversible inhibitors to evaluate the blockage of specific lysosomal proteases and the resultant alteration in APP degradation. They found that APP fragments are routinely generated and then degraded in lysosomal compartments. Furthermore, they found a segregation of protease function between cysteine and non-cysteine proteases. For example, non-cysteine proteases, such as CatD, degrade intact APP to produce potentially amyloidogenic fragments, while cysteine proteases further degrade them [18].

These enzymatic processes can serve as biomarkers for AD and its progression. While much fluorescence-based research has been dedicated to studying brain tumor margins for intraoperative guidance [19] and cellular trafficking of neural progenitor cells [20], few studies have studied enzyme biomarkers responsible for AD. To identify and better understand this enzymatic action, an imaging probe is crucial. Indeed, molecular imaging (MI) of neurodegenerative diseases is critical for understanding and treatment of disease pathology [21,22]. Traditionally, MI probe development has mainly focused on agents using MRI, single-photon emission computed tomography (SPECT), computed tomography (CT), and PET [23,24,25,26] imaging modalities, though optical techniques such as fluorescence imaging are increasingly being utilized and explored [27,28,29].

The work done to develop optical techniques can be used to produce MI probes that can elucidate the roles of BACE1 and CatD in AD pathogenesis. In doing so, BACE1 and CatD can act as enzyme biomarkers and in vivo MI probes can serve as definitive diagnostic tools for AD and its treatment. To reach this goal, we have synthesized a multimodal, multi-wavelength, near-infrared fluorescence (NIRF) probe to detect CatD and BACE1 in cell-free and cell culture conditions based on previous probe design that uses an iron oxide nanoparticle scaffold [19,27,30] to quench fluorescently-labeled enzyme substrates. We believe that this putative AD MI probe has the potential to aid in early AD diagnosis and to speed up the preclinical and clinical assessment of novel AD therapeutic agents.

## 2. Results

The synthesis and fluorescence activation of the AD imaging probe is shown in Figure 1A,B.

The AD MI probe used magneto–fluorescent nanoparticles (cross-linked dextran iron oxide nanoparticles, or CLIO) as a scaffold. The BACE1 substrate contains the sequence surrounding the leucine-asparagine BACE1 cleavage site in APP. In the presence of the CatD and BACE1 proteases, fluorescence activation from the AD MI probe occurs. The AD MI probe consists of two fluorescently-labeled peptide substrates, one for BACE1 and one for CatD, conjugated to an iron oxide core, as shown in Figure 1C. Each fluorescently-labeled substrate has a unique fluorochrome that is detected through UV-VIS spectroscopy: the CatD substrate is labeled with ATTO647N, and the BACE1 substrate is labeled with Cy7. These two peaks are seen in the UV–VIS spectrum for the AD imaging probe and are absent in the spectrum for the bare-amino CLIO (Figure 1C). The iron absorption peak is seen in both spectra.

Enzyme specificity of the AD MI probe for CatD and BACE1 was tested in a 96-well plate assay, as shown in Figure 2.

The fluorescence activation of the AD MI probe using CatD and BACE1 was measured in separate wells (i.e., the enzymes were not mixed in the same well) in a fluorescence microplate reader. Fluorescence was monitored at 669 nm for the BACE1 peptide substrate and at 775 nm for the CatD peptide substrate. Specificity of the BACE1 substrate is shown in Figure 2A, left (red, solid line), illustrating selective cleavage of the BACE1 substrate by BACE1 over CatD (red, dashed line). Compared to CatD activation, this substrate exhibited a 31.5-fold fluorescence increase of BACE1 activation. Shown in Figure 2A (right), the CatD substrate shows selective cleavage by CatD (blue, dashed line) over BACE1 (blue, solid line), demonstrating a 15.5-fold fluorescence increase of the CatD substrate.

Using the conditions described in the Methods section, Michaelis–Menten enzyme kinetic parameters were also examined for both the BACE1 and CatD peptide substrates. The BACE1 substrate was calculated to have a Km value of 11.0 µM and a Vmax value of 12.0 × 10^4^ µM/min. The CatD substrate was calculated to have a Km value of 6.7 µM and a Vmax of 145.2 nM/min. The BACE1 and CatD Lineweaver–Burk transformations had R-squared values of 0.86 and 0.99, respectively.

The imaging ability of the AD MI probe for the detection of CatD and BACE1 was also evaluated in cell culture. H4 and H4 cells stably transfected to overexpress BACE1 (H4-BACE1) cells were utilized. A fluorescence microplate reader was employed to monitor the activation of the AD imaging probe. Western blot analysis of H4 and H4-BACE1 was performed to verify BACE1 and CatD protein levels in these cell lines, as shown in Figure 2B. Compared to H4, H4-BACE1 exhibits higher expression of BACE1 protein as confirmed by Western blot analysis. Incubation of the AD MI probe results in fluorescent signals that correlate well with the expression levels of the BACE1 protein in two cell lines.

CatD protein expression levels in the H4 and H4-BACE1 cell lines were also evaluated and showed slightly higher expression levels of active CatD. The H4 cell lines also show slightly heightened levels of intermediate CatD and pro-CatD. When incubated with both cell lines, the AD imaging probe showed greater fluorescence activation by CatD in the H4-BACE1 cells than for H4 (Figure 2C). These experiments show that the enzyme expression levels correlate to enzyme activity in cell culture. Specifically, the H4-BACE1 cell lines show increased BACE1 and active CatD enzyme levels, indicating increased activity of these enzymes as reflected in the fluorescence activation of the imaging probe.

As shown in Figure 2D,right, the use of a β-secretase inhibitor for the BACE1 substrate suppressed enzyme cleavage of the BACE1 substrate, reflecting a fluorescence increase of the substrate lower than when no inhibitor was used. The decrease in fluorescence signal with the use of an inhibitor was seen in both H4 and H4-BACE1 cells. Similarly, compared to when no inhibitor was used, the use of pepstatin A (CatD inhibitor) also suppressed enzyme cleavage of the CatD substrate, resulting in a lower fluorescence increase (Figure 2D, left). The fluorescence activation fold in H4 and H4-BACE1 cells was lower with the use of pepstatin A than without the use of inhibitors.

## 3. Discussion

In this study, we aim to develop and characterize a novel MI probe that can detect activities of potential AD enzymatic biomarkers- BACE1 and CatD simultaneously. Our current hypothesis is: imaging BACE1 and CatD activities will provide a useful prognostic tool for AD. The rationales are: (1) APP cleavage by BACE1 is the committed step in Aβ amyloidogenesis; (2) CatD has also been shown to cleave APP [31,32] and is upregulated in the lysosomal system in AD specifically [33]. This hypothesis is also based upon our preliminary studies on a novel, multi-wavelength, and potentially multimodal MI probe for simultaneous detection of CatD and BACE1 activities. This CLIO/NIRF-based MI probe has been synthesized and characterized in vitro. This novel MI probe shows the imaging capability to specifically and independently detect both BACE1 and CatD enzymatic activities. The measured activities of BACE1 and CatD in cultured neuronal cells correlated well with BACE1 and CatD protein expression levels, respectively.

Optical imaging probes have been designed and synthesized to monitor and quantitate two distinct enzyme activities in vitro: BACE1 and CatD. The peptide substrates were labeled with fluorochromes whose wavelengths are optically distinct with minimal spectral overlap, or spectral cross talk, between optical channels. Although these probes were not used as such, they have photo properties that would be useful for in vivo imaging. Specifically, the wavelengths of the fluorochromes are in the near-infrared region of the spectrum, allowing for greater photo penetration in tissues, while remaining optically distinct.

Both CatD and BACE1 are transmembrane aspartic proteases, and although the two enzymes are related, BACE1 has a distinct structure from the pepsin family members (which includes cathepsins E and D, pepsinogens A and C, and renin) [34]. Although enzyme specificity between BACE1 and CatD was shown, future experiments include Michaelis–Menten enzyme kinetics studies to compare enzyme specificity between BACE1, CatD, and other aspartic proteases.

The Michaelis parameter (Km) substrate (11.0 µM) is slightly higher than the average Km value of another commercially available fluorogenic peptide substrate for BACE1—11.0 µM for the AD imaging probe and 9.0 for the fluorogenic peptide substrate when tested with BACE1 cloned, expressed, isolated and purified from HEK (human embryonic kidney) cells [35]. This result shows comparable substrate affinity of BACE1 to each of their peptide substrates. The maximal velocity value (Vmax) for the BACE1 substrate is also higher than for the AD imaging probe compared to the aforementioned fluorogenic peptide substrate for BACE1: 12.0 × 10^4^ µM/min compared to 25 nM/min, a significantly higher increase. Higher Km and Vmax values may be attributed to the use of commercially available BACE1 instead of cloning and expressing BACE1 from cells that were used.

In the same study, a fluorogenic substrate for CatD was also used, and the kinetic parameters for the CatD peptide substrate in our AD imaging probe were compared. Commercially available, purified CatD was used in both studies. The CatD peptide substrate had a Km value of 6.7 µM, which is slightly higher than the 2.5 µM value published in [35], indicating a lower affinity for the substrate of our AD imaging probe. The Vmax value of the AD imaging probe was also higher in the AD imaging probe when compared to the published value: 145.2 nM/min compared to 60 nM/min. The difference may be attributed to the sources of commercially available CatD that was used in both assays. The compared reference does not mention the specific species from which the CatD was isolated.

Through the use of H4 and H4-BACE1 cell lines, we monitored in vitro the ability of the AD imaging probe to detect BACE1 and CatD activity. Although the fluorescence activation of the imaging probe reflected the enzyme expression levels in immunoblots, the fluorescence activation also showed non-specific enzyme cleavage by other proteases. Michaelis–Menten enzyme kinetic studies will further clarify non-specific cleavage by proteases other than BACE1 and CatD.

Cell assays were also conducted with inhibitors selective for BACE1 or CatD. Although the AD MI probe still showed some fluorescence activation, both substrates showed lower fluorescence activation with the use of their corresponding inhibitors compared to when no inhibitor was used, revealing the inhibitory effects. The fluorescence activation exhibited in the presence of the inhibitors could be due to other proteases present in the cell culture assay.

In summary, a novel CLIO/NIRF-based molecular imaging probe for the simultaneous detection of CatD and BACE1 activities has been synthesized and characterized in vitro. For the first time, this novel molecular imaging probe specifically and independently detected both BACE1 and CatD enzymatic activities that are believed to be intimately involved in AD pathology. The measured activities of BACE1 and CatD in cultured brain cells correlated well with BACE1 and CatD protein expression levels, respectively. Future studies can include testing the BACE1 substrate specificity and Michaelis–Menten enzyme parameters for CatD and vice versa. In addition, Michaelis–Menten parameters for other aspartic proteases would provide a more complete profile of the behavior of this AD MI probe. Although these imaging probes were not tested for this ability, their iron oxide cores have been used as a magnetic resonance (MR) contrast agent in addition to being part of an optical probe. Future studies are warranted to explore the MR capability of the AD MI probe. This may further validate the roles of these two proteases in AD etiopathogenesis.

## 4. Materials and Methods

### 4.1. Probe Synthesis

#### 4.1.1. Cross-linked Iron Oxide Nanoparticles (CLIO)

Amine-terminated CLIO nanoparticles, consisting of a core of superparamagnetic iron oxide with a cross-linked coating of dextran with amino groups on the surface, were obtained from Dr. Lee Josephson at the Center for Molecular Imaging Research (CMIR) (Boston, MA, USA) General Hospital (MGH).

#### 4.1.2. Peptide Synthesis

Peptides were synthesized by the MGH Peptide Core Facility (Charlestown, MA, USA) on a 100 µmol scale using Fmoc chemistry. Peptide substrates for BACE1 and CatD were synthesized and had the following sequences:BACE1: NH_2_-Glu-Val-Asn-Leu-Asp-Ala-Glu-Phe-Cys-COOHCatD: NH_2_-Pro-Ile-Cys(Et)-Phe-Phe-Arg-Leu-Gly-Cys-COOH

#### 4.1.3. Creation of Fluorescently-labeled Peptide Substrates

With 3 µmol of each peptide, 1 mg of dye in dimethyl sulfoxide (DMSO) and 15 µL of triethylamine were mixed. Covered at room temperature, the reaction sat overnight. The BACE1 peptide was mixed with Cy7 dye (GE Healthcare, Chicago, IL, USA) to produce an excitation wavelength at 749 nm and emission at 775 nm. The CatD peptide was mixed with ATTO647N dye (Sigma-Aldrich, St. Louis, MO, USA) to produce an excitation wavelength at 645 nm and emission at 669 nm.

Using 0.1%TFA (trifluoroacetic acid) in water and acetonitrile with 0.1% TFA as elution buffers on a reversed-phase column (Varian, Palo Alto, CA, USA, 30 × 10 mm), both peptide sequences were purified through high-performance liquid chromatography (HPLC) (Varian). Purity was then verified using analytical HPLC (Varian, 30 × 4.6 mm) and MALDI-TOF (data not shown). The chromatograms for each substrate can be found in Appendix A.

#### 4.1.4. Synthesis of Fluorescently-labeled Nanoparticle Conjugates

A thioether bond was created between the amine-terminated CLIO and the fluorescently-labeled peptide substrates. Amino-CLIO (22 mg Fe, 1.5 mL) was added to 0.5 mL of 150 mM SIA in DMSO and 15 µL of triethylamine. The reaction sat for 1 h at room temperature. Iodoacetyl-CLIO was separated from iodoacetic acid using a Sephadex G-25 column equilibrated with 1× PBS (pH 7.4) at room temperature. We then collected the fluorescently-labeled nanoparticle conjugates.

### 4.2. Measurement of Michaelis–Menten Enzyme Kinetic Parameters

Michaelis-Menten kinetic parameters were examined for the BACE1 and CatD substrates that were conjugated to the iron oxide nanoparticle. Purified BACE1 (Human, Sigma-Aldrich,) and CatD (Human, Calbiochem, San Diego, CA, USA) enzymes were tested separately in 96-well plate formats. To test Michaelis–Menten enzyme kinetics for the BACE1 peptide substrate, the imaging probe was suspended in 20 mM sodium acetate buffer (pH 5) with the following substrate concentrations: 9.3 µM, 4.52 µM, 3.87 µM, and 1.94 µM. For analysis of Michaelis–Menten enzyme kinetics of the CatD substrate, the imaging probe was suspended in 50 mM glycine buffer (pH 3.5) using the following substrate concentrations: 4.5 µM, 3.38 µM, 2.25 µM, and 1.13 µM. Enzyme cleavage was measured as the fluorescence of each peptide substrate increased. Every 5 min, we recorded data points. Fluorescence was monitored at 669 nm for the BACE1 substrate and at 775 nm for the CatD substrate. Measurements were performed on a microplate reader (Molecular Devices, SpectroMax M5e, San Jose, CA, USA). Parameters, such as the Michaelis constant (Km) and maximal velocity (Vmax), were analyzed using non-linear least squares fitting to the Michaelis–Menten equation (through a Lineweaver-Burk transformation) using the SoftMax Pro 2.0 software (Molecular Devices, San Jose, CA, USA).

### 4.3. Fluorescence Activation of the AD Imaging Probe by BACE1 and CatD

The AD imaging probe was tested using purified CatD (Human Liver, Calbiochem, Gibbstown, NJ, USA) and BACE1 (Sigma-Aldrich, St. Louis, MO, USA) in a 96-well plate assay format. To test the fluorescence activation of the nanoparticle imaging probe with CatD, 5–10 µg of the imaging probe was suspended in 50 mM glycine buffer (pH 3.5) with 3 mU of CatD. To test the fluorescence activation of the AD imaging probe with BACE1, 5–10 µg of the imaging probe was suspended in 150 µL of 20 mM sodium acetate buffer (pH 5) buffer containing 10 mM calcium chloride and 10mU of BACE1. The fluorescence activation of CatD and BACE1 as detected by the AD imaging probe was measured in separate wells (i.e., the enzymes were not mixed in the same well) in a fluorescence microplate reader (Tecan, Safire II, Männedorf, Switzerland) at 37 °C for 10 hrs. Fluorescence was monitored at 669 nm for BACE1 substrate activation and 775 nm for fluorescence activation of the CatD peptide substrate. Every 10 min, we recorded data points. We then used Microsoft Excel (Microsoft, Redmond, WA, USA) to analyze the data.

### 4.4. Immunoblotting

Protein concentrations of cell lines were measured using a BCA protein assay kit from Pierce (Rockford, IL, USA). Equal amounts (10 µg of protein) of cell lysate from H4 or H4-BACE1 cells were heated at 70 °C for 15 min in the SDS sample loading buffer before being separated on a 4–12% NuPAGE Bis-Tris gel in MES (2-(N-morpholino) ethanesulfonic acid) buffer (Invitrogen, Carlsbad, CA, USA) and then transferred to a nitrocellulose membrane (Invitrogen, Carlsbad, CA, USA). The blots were blocked at RT in 5% nonfat dry milk (Fisher Scientific, Hampton, NH, USA) made in Tris-buffered saline with 0.1% Tween 20 (TBS-T) (Sigma-Aldrich, St. Louis, MO, USA) and then incubated with commercial CatD (CTD-19) antibody (Abcam, Cambridge, MA, USA) diluted in 5% nonfat milk in TBS-T at the concentration recommended by the manufacturer. Blots were then washed with TBS-T and incubated with HRP-conjugated goat anti-mouse secondary antibody (1:10,000; 1h at RT). Blots were then developed for use with a SuperSignal West Pico Chemiluminescent Substrate (ECL) (Pierce), and immunosignals were captured using x-ray film (ThermoFisherScientific, Waltham, MA, USA). The blots were then stripped with stripping buffer (Restore Western Blot Stripping Buffer, ThermoScientific), re-probed with BACE1 antibody (ab2077) (Abcam, Cambridge, MA), and incubated with HRP (horseradish peroxidase)-conjugated goat anti-rabbit secondary antibody (1:10,000; 1h at RT). We exposed the film again. Once re-stripped, the films were re-probed with β-actin antibody for standardization.

### 4.5. Fluorescence Activation of the AD Imaging Probes in H4 and H4-BACE1 Cell Assays

The AD imaging probe was tested in cell culture using the human H4 neuroglioma cell line and H4 cells stably transfected to overproduce BACE1 (H4-BACE1). H4 cells were cultured in Dulbecco’s modified eagle medium (DMEM) containing 10% fetal bovine serum (FBS) (Invitrogen, Carlsbad, CA, USA) and 1% penicillin/streptomycin in a black 96-well plate with a clear glass coverslip bottom (Corning Costar, Corning, NY, USA). H4-BACE1 was cultured in a DMEM medium containing 10% FBS, 1% L-glut/penicillin/streptomycin (Sigma-Aldrich, St. Louis, MO, USA), and 0.5% G418 (EMD Biosciences, San Diego, CA, USA). After 24 hrs, the cell culture medium was replaced with 200 µL of fresh medium containing 5–10 µg of the AD imaging probe and monitored in a fluorescence microplate reader (Safire II, Tecan, Männedorf, Switzerland) at 37 °C with data points taken every 10 min. We used Microsoft Excel to acquire and analyze the data points.

### 4.6. Inhibition of BACE1 and CatD Enzyme Activities in Cell Assays

Substrate specificity and activity in cell culture was also measured using inhibitors specific for BACE1 and CatD. The AD imaging probe was tested in cell culture using H4 and H4-BACE1 cells lines in conditions similar to those described above. Inhibition of the BACE1 or CatD substrate were tested separately using inhibitors for BACE1 (β-secretase inhibitor, 1 mg/µL, Sigma-Aldrich, St. Louis, MO, USA) or CatD (pepstatin A, Sigma-Aldrich, 1 mg/µL).

## Figures and Tables

**Figure 1 molecules-25-00274-f001:**
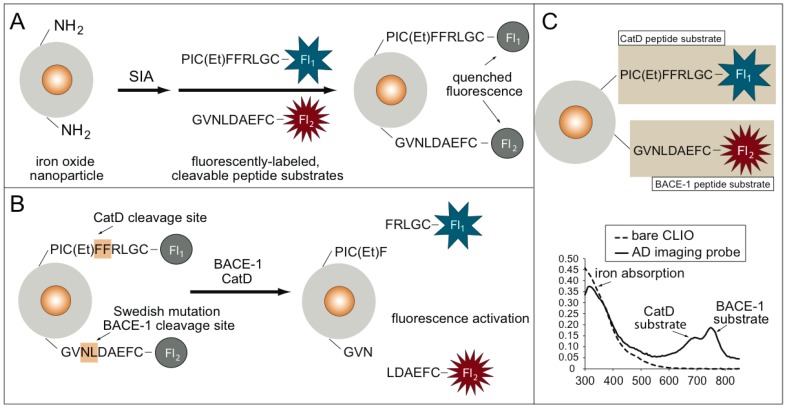
(**A**) Synthesis of the Alzheimer’s disease (AD) imaging probe using an iron oxide nanoparticle scaffold and fluorescently-labeled peptide substrates for cathepsin D (CatD) and cleaving enzyme 1 (BACE1) AD enzymatic biomarkers. (**B**) Fluorescence activation of the AD imaging probe. Fluorescence is quenched until either the presence of CatD or BACE, which cleaves the fluorescence substrates, results in a fluorescence signal increase. (**C**) UV–VIS spectrum data confirm the synthesis of the AD molecular imaging (MI) probe.

**Figure 2 molecules-25-00274-f002:**
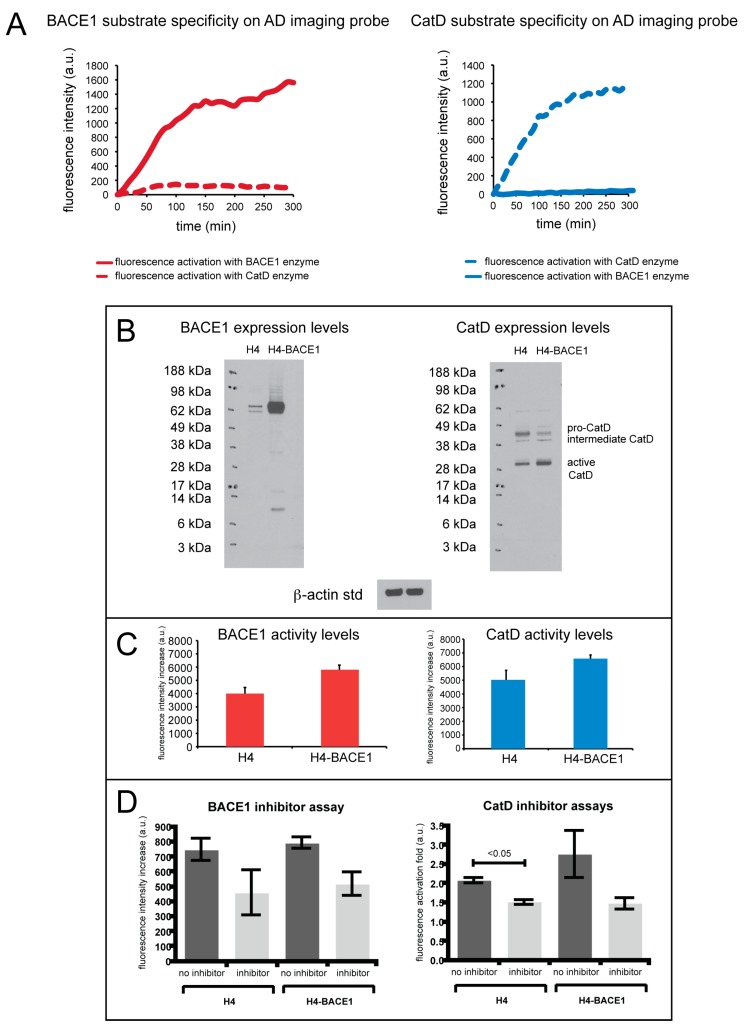
(**A**) (left) BACE1 substrate specificity with BACE1 (solid line) and CatD (dashed line) enzymes on the AD imaging probe show that the BACE1 peptide substrate is selectively cleaved when compared to CatD. (right) CatD substrate specificity with BACE1 (solid line) and CatD (dashed line) enzymes on the AD imaging probe, illustrating selective cleavage of the CatD substrate by CatD when compared to BACE1. (**B**) Western blot analysis of native H4 cell lines (H4) and BACE1 overexpressing H4 cell lines (H4-BACE1). (**C**) Fluorescence assay of H4 and H4-BACE1 cells incubated with AD imaging probe measuring BACE1 enzyme activity (right) and CatD enzyme activity (left). (**D**) Fluorescence assay of H4 and H4-BACE1 cells incubated with AD imaging probe with inhibitors for BACE1 and CatD. The graph on the right measures BACE1 activity in H4 and H4-BACE1 cells with and without inhibitors specific for BACE1, and the graph on the left measures CatD activity in H4 and H4-BACE1 cells with and without inhibitors specific for CatD. The error bars reflect the SEM of the average values.

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
