# Peer review of "A Novel Dual Fluorochrome Near-Infrared Imaging Probe for Potential Alzheimer’s Enzyme Biomarkers-BACE1 and Cathepsin D"

_molecules, 2020, doi:10.3390/molecules25020274_

Round 1

Reviewer 1 Report

In this contribution, the authors described a molecular imaging probe to fluorescently image the β-site of the amyloid precursor protein (APP) cleaving enzyme 1 (BACE1) and cathepsin D (CatD) enzymes associated with Alzheimer’s disease. The results show that when incubated with purified BACE1 enzyme, the CatD peptide substrate demonstrated a 15.5-fold increase in fluorescence when incubated with purified CatD enzyme, and the BACE1 substrate exhibited a 31.5-fold increase in fluorescence. There are some points that need to be addressed before the paper will be ready for publication.

What about the size changes of the core material, iron oxide nanoparticles, in the processes of fluorescence labelling and fluorescence activation? And the cell viability of the fluorescently-labeled nanoparticle conjugates. Table 2 is not readable because of low clarity. Is there any relationship between the loading efficiency and the fluorescence enhancement of the substrate when incubated with purified enzyme? Please check the reference style of the Ref 4 and ref 35, and also the tittle style of all the references.

Author Response

Reviewer 1 critiques

“What about the size changes of the core material, iron oxide nanoparticles, in the processes of fluorescence labelling and fluorescence activation? And the cell viability of the fluorescently-labeled nanoparticle conjugates.”

Response: As this work was meant to serve as a proof-of-concept for the particle design, we did not check size changes or cell viability in the current studies. We will perform the experiments in the future studies. Thanks for your suggestion.

“Table 2 is not readable because of low clarity.”

Response: The reviewer means Figure 2. We have replaced it with new Figure 2.

“Is there any relationship between the loading efficiency and the fluorescence enhancement of the substrate when incubated with purified enzyme?”

Response: Unfortunately, we did not calculate the loading/encapsulation efficiency in this proof-of-concept study. We will do it in the future studies. Thanks for your suggestion.

“Please check the reference style of the Ref 4 and ref 35, and also the tittle style of all the references.”

Response: Yes, we have made the corrections by re-formatting the references using “Molecules” journal Endnote style.

Reviewer 2 Report

The manuscript entitled “A novel dual fluorochrome near-infrared imaging probe for potential Alzheimer’s enzyme biomarkers-BACE1 and cathepsin D” by Huang and co-workers is well written and describes in detail the synthesis and the activity profile of this new imaging probe. The authors idea of creating such a molecular probe, in order to identify biomarkers related to AD and ADRD, in a simpler and cheaper way, is well described and carried out.

In particular the imaging probe is built upon iron oxide nanoparticles and linked to peptidic sequences containing a near-infrared fluorochrome, giving the possibility of monitoring the activity of CatD and BACE1 as biomarkers for AD. The imaging ability of the probes in detecting CatD and BACE1 was also evaluated in H4 and H4-BACE1 cell cultures.  The carried out experiments support the application of this NIRF in the early AD diagnosis and the opportunity of accelerating the clinical and preclinical studies of agents for AD and ADRD.

I suggest some minor revisions before publication:

The spelled-out form of some acronyms is missed and has to be added;

The quality of figure 2 has to be improved;

The higher Km value detected for CatD substrate does not indicate a higher affinity compared to data reported in reference 35, please clarify this aspect;

Concerning the experiment carried out in the presence of the two selective inhibitors some information are missed. Is it possible to determine the percentage of inhibition reached by each inhibitor during the experiment?

Line 189 the unit of measurement used for Km value have to be corrected;

Line 194/195 the reported Vmax value for the BACE1 substrate is different from the one reported on line 141;  

Line 274 the unit of measurement for CatD units is not correct in this format.

Author Response

Reviewer 2 critiques

“The spelled-out form of some acronyms is missed and has to be added;”

Response: We have spelled out the remaining acronyms/abbreviations, and have added them to the list of abbreviations.

“The quality of figure 2 has to be improved;”

Response: Yes, we have replaced it with high quality Figure 2.

“The higher Km value detected for CatD substrate does not indicate a higher affinity compared to data reported in reference 35, please clarify this aspect;”

Response: You are right, we had the relationship written out backwards. The higher Km of our substrate is indicative of a lower relative affinity.

“Concerning the experiment carried out in the presence of the two selective inhibitors some information are missed. Is it possible to determine the percentage of inhibition reached by each inhibitor during the experiment?”

Response: There is precedent for using the %change in the fluorescence increases observed between the inhibited and uninhibited samples as an approximation for the percent inhibition. In our case, the goal was to show the presence of inhibition, rather than the magnitude of that inhibition.

“Line 189 the unit of measurement used for Km value have to be corrected;”

Response: We have corrected the invalid symbol, thank you for pointing this out.

“Line 194/195 the reported Vmax value for the BACE1 substrate is different from the one reported on line 141;“ 

Response: It is a typo, and we have corrected it.

“Line 274 the unit of measurement for CatD units is not correct in this format.”

Response: You are right, we have corrected the typo.

Reviewer 3 Report

Dear Editor. The authors submitted the Ms “A novel dual fluorochrome near-infrared imaging probe for potential Alzheimer’s enzyme biomarkers- BACE1 and cathepsin D4. By Jenny M. Tam, Lee Josephson, Alexander R. Pilozzi, and Xudong Huang” for possible publication at the Journal. In Introduction Say that there is an urgent and unmet medical need for imaging probes of biomarkers that can reliably distinguish normal from abnormal brain function or cognition and robustly predict or correlate with its clinical decline. They long-term goal is to develop novel molecular imaging probes of relevant biomarkers for characterizing, diagnosing, and predicting outcomes in AD and ADRD. AD diagnosis currently relies on behavior-based tests that are not specific for AD. Evidence suggests that amyloid-beta (Aβ) may be a key step during AD progression. This Aβ peptide, aggregate into plaques, and produced after the sequential cleavage of two proteases, called β- and γ-secretase, that mediate the endoproteolysis of amyloid precursor protein (APP), a type I membrane protein [9]. Thus, β-secretase cleavage is the committed step in Aβ amyloidogenesis and BACE1 is considered as the major form of β-secretases. As such, it has become one of the therapeutic targets for AD. Cathepsin D (CatD), are a part of the lysosomal system, and part of the dysfunction involved in AD. Lysosomal acidification and normal proteolytic activity are found to be compromised in Alzheimer’s disease and other diseases of the central nervous system. Cathepsins, affect the production and removal of intracellular Aβ, are upregulated, disrupting the lysosomal system to ultimately increase intracellular Aβ levels to a point at which Aβ is secreted extracellularly as Aβ aggregates. CatD play a role in processing APP to form Aβ. Siman et. al. used several irreversible inhibitors to evaluate the blockage of specific lysosomal proteases, resulted alteration in APP degradation. Thus the APP fragments are generated and degraded in lysosomal compartments. The results indicate, that non-cysteine proteases - CatD, degrade intact APP to produce amyloidogenic fragments, while cysteine proteases degrade the fragments. The authors developed optical techniques to produce MI probes that can elucidate the roles of BACE1 and CatD in AD pathogenesis. The authors used the APP cleavage by BACE1 and CatD as enzymes biomarkers. The in vivo MI probes can serve as diagnostic tools for AD and its treatment. They aim was synthesizing a multimodal, multi-wavelength, near infrared fluorescence (NIRF) probe to detect CatD and BACE1 in cell-free and cell culture conditions based on previous probe design that uses an iron oxide nanoparticle scaffold to quench fluorescently-labeled enzyme substrates. They propose that this putative AD MI probe has the potential to show the imaging capability to specifically and independently detect both BACE1 and CatD enzymatic activities in order to aid in early AD diagnosis and to speed up the preclinical and clinical assessment of novel AD therapeutic agents. Therefore, a novel CLIO/NIRF-based molecular imaging probe for the simultaneous detection of CatD and BACE1 activities has been synthesized and characterized in vitro. For the first time, this novel molecular imaging probe specifically and independently detected both BACE1 and CatD enzymatic activities that are believed to be intimately involved in AD pathology. The measured activities of BACE1 and CatD in cultured brain cells correlated well with BACE1 and CatD protein expression levels, respectively. The results were well written, well discussed, with good references, and appropriate abstract. Therefore I suggest accepting for publication at the Journal

Author Response

Reviewer 3 Critiques

“…The results were well written, well discussed, with good references, and appropriate abstract. Therefore I suggest accepting for publication at the Journal.”

Response: We are very grateful for this reviewer’s very encouraging comments upon our manuscript.

Round 2

Reviewer 1 Report

  This manuscript can be accepted in present form.